# Space Environmental Chamber for Planetary Studies

**DOI:** 10.3390/s20143996

**Published:** 2020-07-18

**Authors:** Abhilash Vakkada Ramachandran, Miracle Israel Nazarious, Thasshwin Mathanlal, María-Paz Zorzano, Javier Martín-Torres

**Affiliations:** 1Department of Computer Science, Electrical and Space Engineering, Luleå University of Technology, 97187 Luleå, Sweden; miracle.israel.nazarious@ltu.se (M.I.N.); thasshwin.mathanlal@ltu.se (T.M.); maria-paz.zorzano.mier@ltu.se (M.-P.Z.); javier.martin-torres@ltu.se (J.M.-T.); 2Centro de Astrobiología (CSIC-INTA), Torrejón de Ardoz, 28850 Madrid, Spain; 3School of Geosciences, University of Aberdeen, Aberdeen, AB24 3FX, UK; 4Instituto Andaluz de Ciencias de la Tierra (CSIC-UGR), 18100 Granada, Spain

**Keywords:** space, environmental chamber, Mars simulation, vacuum, planetary atmosphere, space instrumentation

## Abstract

We describe a versatile simulation chamber that operates under representative space conditions (pressures from < 10^−5^ mbar to ambient and temperatures from 163 to 423 K), the SpaceQ chamber. This chamber allows to test instrumentation, procedures, and materials and evaluate their performance when exposed to outgassing, thermal vacuum, low temperatures, baking, dry heat microbial reduction (DHMR) sterilization protocols, and water. The SpaceQ is a cubical stainless-steel chamber of 27,000 cm^3^ with a door of aluminum. The chamber has a table which can be cooled using liquid nitrogen. The chamber walls can be heated (for outgassing, thermal vacuum, or dry heat applications) using an outer jacket. The chamber walls include two viewports and 12 utility ports (KF, CF, and Swagelok connectors). It has sensors for temperature, relative humidity, and pressure, a UV–VIS–NIR spectrometer, a UV irradiation lamp that operates within the chamber as well as a stainless-steel syringe for water vapor injection, and USB, DB-25 ports to read the data from the instruments while being tested inside. This facility has been specifically designed for investigating the effect of water on the Martian surface. The core novelties of this chamber are: (1) its ability to simulate the Martian near-surface water cycle by injecting water multiple times into the chamber through a syringe which allows to control and monitor precisely the initial relative humidity inside with a sensor that can operate from vacuum to Martian pressures and (2) the availability of a high-intensity UV lamp, operating from vacuum to Martian pressures, within the chamber, which can be used to test material curation, the role of the production of atmospheric radicals, and the degradation of certain products like polymers and organics. For illustration, here we present some applications of the SpaceQ chamber at simulated Martian conditions with and without atmospheric water to (i) calibrate the ground temperature sensor of the Engineering Qualification Model of HABIT (HabitAbility: Brines, Irradiation and Temperature) instrument, which is a part of ExoMars 2022 mission. These tests demonstrate that the overall accuracy of the temperature retrieval at a temperature between −50 and 10 °C is within 1.3 °C and (ii) investigate the curation of composite materials of Martian soil simulant and binders, with added water, under Martian surface conditions under dry and humid conditions. Our studies have demonstrated that the regolith, when mixed with super absorbent polymer (SAP), water, and binders exposed to Martian conditions, can form a solid block and retain more than 80% of the added water, which may be of interest to screen radiation while maintaining a low weight.

## 1. Introduction 

The interest in designing instrumentation for the exploration of the Moon or Mars has increased over the past few years, and various space agencies and companies have demonstrated it. The hardware used in these missions must be capable of operating under extreme environments. In order to facilitate the procedures and phases of instrumentation mainly to design, qualify, and calibrate, we have designed the SpaceQ chamber. The primary purpose of this chamber is to test sensors and components when exposed to representative space conditions with thermal or pressure changes. It is also used to recreate conditions of the Martian or Lunar surface.

One specific capability of this chamber is that it allows simulating the water cycle and environmental conditions that will be experienced by the HABIT (HabitAbility: Brines, Irradiation and Temperature) instrument of the ExoMars 2022 mission that will land at Oxia Planum and operate from the Martian surface. Pretesting under representative space conditions is a requirement to raise the technological readiness level (TRL) of any instrument proposed to a space mission. HABIT will investigate the potential present-day habitability at Oxia Planum, Mars, examining the ground and air temperature, the electric conductivity of the air, ultraviolet (UV) radiation, wind and liquid, and brine formation by deliquescence of a set of salts [1]. It will operate autonomously through day and night, acquiring environmental observations for, at least one Martian year of nominal operation of the Surface Platform *“Kazachok”.*

Since the 1960s, there have been numerous developments of simulation chambers around the world, including the chamber to study the behavior of terrestrial microorganisms under artificial Martian conditions [2] and other multiple simulating facilities for planetary and space research [3,4,5,6,7,8,9,10,11,12,13,14,15,16,17]. The design of this kind of facilities is still an active area of interest, as each one is generally built with a few specific objectives. For instance, just recently, a Mars environmental chamber has been developed to allow dust to remain in suspension and to calibrate an optical particle counter [18]. Due to the need for simulating the Martian water cycle to test the HABIT instrument and other processes that depend on moisture, we have developed the SpaceQ chamber. The core novelties of this chamber are: (1) its ability to simulate the Martian near-surface water cycle by injecting water multiple times into the chamber through a syringe which allows to control, precisely, the initial relative humidity inside and (2) the availability of a high-intensity UV lamp operating within the chamber, which can be used to test material curation, the role of the production of atmospheric radicals, and the degradation of certain products like polymers and organics. This chamber can test scientific instruments and, since water is a requirement for life as we know it, this chamber will allow to perform experiments related to astrobiology and habitability. 

Mars has a thin atmosphere, which consists mostly of carbon dioxide (CO_2_), and water is only a minor component. However, due to the large diurnal thermal oscillations, the surface can go from 0 to 100% relative humidity (RH) within 1 day [19]. Water is not stable as a liquid on Mars, but it can exist as frost and ice in the ground, and in the atmosphere as vapor, fogs, and cloud nuclei. Moreover, water can also be bonded or adsorbed to salts in the regolith. In this state, there can be an interchange of water between the regolith and the atmosphere under present-day conditions due to the diurnal and seasonal variations of temperature and relative humidity. Then, additionally, water can be fixed in the mineral crystal structure, or as hydroxyl, but this water is only released at temperatures above 200 °C and up to 1000 °C or higher, depending on the mineral. 

Some of the salts that exist on Mars may even deliquesce and form liquid brines that are stable at these pressures and temperatures. The presence of salts on Mars has implications on the water circulation and water cycle as well as on life and materials and instruments exposed to these surface environmental conditions. This chamber allows testing the performance of certain materials. Some metallic materials may be damaged when they come in contact with brines, as they are usually very corrosive. Other materials may hydrate or dehydrate differently as on Earth, so this chamber allows to test the properties of certain materials during curation under Martian conditions. Moreover, finally, since liquid water is needed for life, the formation of brines may have implications on life and planetary protection protocols. In the Experimental Procedure section, we present some examples of the applications of the SpaceQ chamber. 

## 2. SpaceQ Detailed Description

The SpaceQ is a stainless-steel cubic-shaped (Figure 1) customized chamber manufactured by Kurt J. Lesker Company (https://www.lesker.com/index.cfm). The door is made of aluminum, and the internal volume is 27,000 cm^3^. The quartz silica viewports (Zero-Length, 4-1/2” UHV) fitted on the top and left-hand side allow to monitor the experiment inside. The glass window is of Kodial with a viewing diameter of 65 mm. This glass is compatible with high vacuum and can withstand a bake-out temperature of up to 623 K. The heating jackets are custom made by Eltherm (https://eltherm.com/). The outer cover of the jackets is made of glass fabric PTFE coating, and the inner cover is of glass fabric. The thermal insulation is made of 15 mm glass wool and uses Velcro fasteners as fixing material around the chamber. The jackets are fitted to the external walls of the chamber and have an operating temperature from ambient to 423 K. It has platinum (PT) 100 as its sensing element with 700 W nominal output to supply heat through the walls. The temperature is set using an electronic temperature controller (ELTC)-15 with an output relay and a programmable ramp. This feature is used for outgassing tests and to convert the chamber into a thermal vacuum test (TVT) chamber where instruments can be tested and thermally cycled to perform space qualification and investigate thermal balance and gradients within the hardware. The chamber wall is fitted with 12 utility ports (KF, CF, Swagelok) on which different instruments are fitted to perform the tests inside. It has sensors for temperature (T), relative humidity (RH%), and pressure (P), an ultraviolet (UV) irradiation lamp, an ultraviolet (UV)–visible (VIS)–near infrared (NIR) spectrometer, as well as a stainless-steel syringe for water vapor injection, and universal serial bus (USB) and distribution board (DB) 25 ports to read the data from the instruments. A functional diagram of SpaceQ is shown in Figure 2.

SpaceQ is very versatile in terms of its operating temperatures; it can operate in the temperature range of 163–423 K and the pressure range from < 10^−5^ mbar (high vacuum) to ambient pressure equipped with different sensors. As the SpaceQ chamber can operate up to 423 K, through an external heating jacket, it can be used to qualify components for space operations by performing TVT (ECSS-Q-ST-70-02C), outgassing, and sterilization through dry heat microbial reduction (DMHR) tests (ECSS-Q-ST-70-57C). The SpaceQ has been used to simulate the Martian environment and the water and thermal cycle to investigate the dew point, the frost formation point, and the formation and stability of brines [20]. It can be used to perform microbiological experiments and test the curation of different materials that have been so far only tested under Earth ambient conditions [21]. The specifications of the chamber are summarized in Table 1. 

The dimensions of the cooling plate (Figure 3) are 200 mm × 200 mm × 18 mm, and it has an inbuilt thin pipe for liquid nitrogen (LN_2_). The temperature of the surface is controlled by passing liquid nitrogen from a Dewar using a feedthrough. The temperature control is performed by a proportional integral differential (PID) controller, which is in a feedback through a PT100 sensor allowing the solenoid valve to open and permit the flow of liquid nitrogen. We monitor the temperature of the working plate connected to a K-type thermocouple located on the left wall of the chamber. The temperature of the samples inside the chamber can be reduced up to 163 K. 

Phase DUO 10 M rotary pump allows to reach from ambient to 10^−3^ mbar, and a HiPace 80 turbo molecular pump can reach up to 10^−5^ mbar. The pressure is controlled by a Cold Cathode/Pirani Combination Gauge, covering a range from 1000 to 1 × 10^−9^ mbar.

Martian or planetary atmospheres are investigated by injecting gas from an external pressurized bottle. For simplicity, the initial testing is conducted with 100% CO_2_ to simulate the Martian atmosphere. The chamber has the flexibility to simulate different gas compositions and pressures. Martian pressure conditions were simulated by first evacuating the chamber and then flushing with CO_2_ until it reaches 6–7 mbar. To inject trace amounts of water, we use a KD Scientific stainless-steel syringe from Fischer Scientific. It has a capacity of 20 mL and is fitted on a Swagelok ¼″ connector, which is in turn connected through a tube to a manually operated ¼″ Swagelok fitting ball valve. The smallest amount that can be injected is 0.5 mL, and this allows us to have precise control on the initial relative humidity conditions. The water can be injected several times during the experiment. When the ball valve is open, the water is sucked into the vacuum or reduced pressure environment. 

To measure the atmospheric temperature and relative humidity inside the chamber, a Vaisala HMT334 sensor is used [22]. This sensor can operate from Martian pressures to vacuum conditions. A similar sensor is operating on Mars since 2012, as part of the Rover Environmental Monitoring Station (REMS) instrument, onboard the Curiosity rover [23]. It can measure the temperature in the range of 203–453 K and the relative humidity from 0 to 100%. This is fitted on the chamber through an M22 × 1.5 thread with the probe exposed to the inner atmosphere which is tested for vacuum-tight installations. Apart from the Vaisala sensor, the chamber has two type K thermocouple feedthroughs (top and left side) which can be used to connect PT 1000 sensors to measure temperatures at any location inside the chamber required by the experiment. The thermocouple wires are made of Chromel (positive wire) and Alumel (negative wire). The feedthrough is a KF 16 flange.

A Hamamatsu S2D2 VUV lamp with output stability fluctuation 0.05% p-p (maximum) can be used, operating directly within the chamber, to simulate the high UV irradiance levels that are experienced in space and on Mars. It can also be used to sterilize the chamber for specific experiments, to induce photodissociation of molecules in the gaseous atmosphere and to test the degradation of some materials such as polymers when exposed to UV and Martian atmospheric conditions. This lamp irradiates within the spectral range of 115–400 nm. It can be incorporated on the top flange of the SpaceQ chamber. It uses a deuterium lamp with a magnesium fluoride (MgF_2_) window fitted with a flexible tube allowing to irradiate objects and samples at a very close distance to be operated under depressurized conditions. UV, visible, and infrared reflectometry studies of samples can be performed by combining two spectrometers (AvaSpec-ULS2048 LTEC-USB2-RS and AvaSpec-NIR256-2.5-HSC-EVO) covering UV–VIS–NIR range with 200–2500 nm. It uses a mini-halogen lamp as a light source to irradiate the sample and measure the reflectance. It is fitted onto the chamber using two vacuum feedthroughs SMA-905, which have an M12 thread housing with Viton O rings designed for the fiber optics in vacuum chambers that interconnects to couple with the probes inside.

## 3. Martian Near-Surface Water Cycle and Thermal Studies

We simulate the environmental conditions that occur on Mars during the transition from nighttime (when due to the low temperature the surface humidity reaches its highest value and may saturate) to daytime (when the surface temperature increases releasing the water to the atmosphere). The simulation is done by evacuating the air in the SpaceQ chamber with a rotary pump down to 10^−3^ mbar and replacing it by carbon di oxide (CO_2_) gas to 6–8 mbar, which is a representative range of the average atmospheric pressure on Mars. A Pirani gauge reader monitors this. Once the CO_2_ gas stabilizes, we inject water using the Swagelok stainless steel syringe and then use liquid nitrogen (LN_2_) to cool the working table down to 250.15 K. This whole process of achieving close to Martian condition takes about 1 h, and the tests are done with pure CO_2_ gas and minute amounts of water. When water is injected, the relative humidity within the chamber changes depending on the temperature. There are unavoidable thermal gradients, which occurs when the table is refrigerated with LN_2_, and the external walls of the chamber are warm due to the contact with the ambient laboratory temperatures. For this specific setup, the relative humidity (RH) probe monitors the air relative humidity with respect to liquid at roughly 10.2 cm above the table and 5.2 cm from the sidewalls. Figure 4 shows an example of a simulation from a Martian night-to-day transition. The ground temperature (T_g_) and air temperature (T_a_), the relative humidity of the air (RH_a_), and pressure (P) are directly measured, whereas the relative humidity of ground with respect to ice and liquid (RH_g_^i^ and RH_g_^l^) are derived by using the formulae (1)–(4) [19].
(1)$$RH\;\left(T\right)=\frac{P}{ew_{\left(T\right)}}\times \frac{vmr}{1+vmr}\times 100$$
(2)$$ew_{liq\left(T\right)}=6.112\times e\left(17.62\times \frac{T-273.14159}{243.12+\left(T-273.14159\right)}\right)$$
(3)$$ew_{ice\left(T\right)}=6.112\times e\left(22.5\times 1-\frac{273.14159}{T}\right)$$
(4)$$vmr=\frac{W}{1000}\times \frac{M_{d}}{M_{w}}$$
where RH(T_a_) can be applied to the air, to retrieve *vmr* (volume mixing ratio) and then be used to calculate RH_g_ (T_g_). Here, RH^i^ represents the relative humidity with respect to ice, whereas RH^l^ represents the relative humidity with respect to liquid.

RH: relative humidity in % P: Pressure in mbarvmr: volume mixing ratio in parts per millionT_g_: table temperature in K$ew_{liq\left(Tg\right)}$: saturation partial pressure over liquid water at a given temperature$ew_{ice\left(Tg\right)}$: saturation partial pressure over ice at a given temperatureM_w_ = 18.0160 (molecular weight of water)M_d_ = 43.3400 (molecular weight of dry air on Mars)W = water mass mixing ratio.

In the example of Figure 4, at 12:00 p.m., the temperature of the table is reduced to 240 K. This is named Tg in the graph, as it represents the nighttime ground temperature at a Martian location. LN_2_ is then stopped, and the temperature increases due to the thermal equilibration with the ambient laboratory temperature, which is around 290 K. As the table temperature is cold, the water vapor in the atmosphere condenses on the table and may even permit the formation of frost when the RH_g_^i^ is at or above saturation. At the first moment, the ambient air humidity (RH_a_) is close to zero as all the water condenses on the cold surface. As the table temperature increases slowly, this water is released back to the atmosphere, which in turn produces an increase of the chamber pressure P, while also raising RH_a_. The pressure is maintained within 8 mbar by manipulating the valve. 

This kind of simulated near-surface water cycle environment conditions on Mars can be used to test instruments for In Situ Resource Utilization (ISRU), curation of materials for construction, testing materials for corrosiveness, to study the stability of liquid water or brines, and to investigate the limits of microbial life under Martian conditions. 

## 4. Material Testing Studies

The plans of deep-space human exploration require the demonstration of In Situ Resource Utilization (ISRU) possibilities for the Martian and lunar surfaces. It has been suggested that the regolith, available on the Moon and Mars, can be used as critical foundational material for the construction of structures, which will provide effective shielding against the high-energy radiation bombarded from the outer space [24,25,26,27]. To be prepared for the future exploration of Mars, it is important to first demonstrate and prove the method for curation in a small-scale simulating conditions here on Earth. This will reduce the cost of transportation of materials from Earth. This is one of the applications of the SpaceQ.

The main goal of the experiment that follows is to demonstrate, the application of the SpaceQ chamber to cure under Martian conditions mixtures of Martian soil simulants when mixed with composite products at different concentrations. These kinds of products have been cured previously at ambient Earth conditions [28,29,30,31,32], but this chamber allows to test the effect of temperature, pressure, UV, and atmospheric moisture under Martian conditions. The purpose of this test is to demonstrate, for the first time under Martian conditions, the fabrication of a solid block with potentially enhanced radiation screening properties due to the presence of water. To enhance the water retention capability of a block, we have also tested the addition of other product, i.e., a super absorbent polymer (SAP), which can hold up to 150–300 times its weight in water in a jelly state [33,34]. For some tests, we have used ferric sulfate (Fe_2_(SO_4_)_3_), which is a salt that exists on the surface of Mars and has deliquescent properties. We hypothesize that if this kind of products can be incorporated into construction elements, this would reduce the weight of structures and still have strong radiation protection properties because of the high content of water [35]. In addition, by using these water-absorbing elements within the composite mixture at Martian conditions, we force a slower dehydration, which may be interesting to emulate the slow curation process on Earth.

### 4.1. Materials and Methods

To simulate the Martian soil, we have used Mojave Martian Simulant (MMS-1) unsorted grade obtained from The Martian garden, Texas, with the composition of high-quality iron-rich basalt mostly used for research purposes [36]. The binder used in this study was EPON 828 epoxy resin purchased from Polysciences because of its excellent mechanical properties and ease of mixing with the regolith. It also has great radiation-screening properties, and many previous studies have reported the use of epoxy resins as binders with the simulants. The hardener used was m-Xylylenediamine from Sigma Aldrich. Ferric sulfate Fe_2_(SO_4_)_3_ and SAP (poly(acrylamide-co-acrylic acid)) (C_6_H_8_KNO_3_) were purchased from Sigma Aldrich. The samples were prepared in a glass beaker. 

In these tests, we prepared two sets of samples under laboratory conditions. One was then placed inside SpaceQ and the other one left at ambient conditions on a laboratory bench. The sample inside SpaceQ was frozen by cooling the table to 258 K with the LN_2_, to prevent the semiliquid mixture from rapid sublimation when the chamber is vacuumed. Once the chamber was vacuumed, we injected CO_2_ and let the temperature rise to ambient Earth temperatures by reaching equilibrium with the laboratory environment. For this test, the atmosphere was dry, so the wet sample released water to the atmosphere during the curation process. Although the sample dries up, the relative humidity (RH) at the air above increases. We consider that the curation has ended when the relative humidity in the chamber is stabilized. 

### 4.2. Results 

For the study, we prepared two different samples (i) mixture of soil, salt, and binder and (ii) mixture of soil, salt, a binder with the addition of SAP and water.

(i): For this test, we used small amounts of samples spread over a large area and prepared five different samples with increasing soil to binder ratio:(a)1 g soil + 0.1 g Fe_2_(SO_4_)_3_ + 0.2 g binder(b)1.5 g soil + 0.15 g Fe_2_(SO_4_)_3_ + 0.2 g binder (exposure to UV)(c)1.5 g soil + 0.15 g Fe_2_(SO_4_)_3_ + 0.2 g binder(d)2 g soil + 0.15 g Fe_2_(SO_4_)_3_ + 0.2 g binder(e)2.5 g soil + 0.15 g Fe_2_(SO_4_)_3_ + 0.2 g binder

The samples were prepared in large beakers, spread over the bottom crystal, placed inside the SpaceQ, and covered with a high efficient particulate air (HEPA) filter to protect the chamber from potential micron sized splashes of the prepared mixture, which may be released abruptly during the vacuum phase. For this test, sample (b), which was equal to (a), was irradiated with a UV lamp fitted on the chamber to check if the resin was damaged upon UV exposure. The UV lamp was switched off after 1 hour(h) of irradiation. As the table temperature rose, the RH stabilized, and the experiment was stopped. The final sample mixtures are shown in Figure 5. Only the samples with a higher ratio of binder to soil were able to form conglomerate (a, b, and c). No significant difference was observed due to UV exposure (b). 

We demonstrated that Martian simulants, when mixed with binders and salts at different concentrations, could be cured under a simulated dry Martian environment. In the study (i), we found that the samples with a higher binder to soil ratio (Figure 5a–c) were able to mix and form solid grains unlike in Figure 5d,e which was quite loose and dispersed. In addition to the granular binding properties, it is essential to mold the material into a solid unit. So, next (ii) we used a smaller beaker for the mixture to form a solid block. 

(ii): In this test, we used SAP, as an extra additive, and higher amounts of soil and binder to make a solid matrix. We used a ratio of 30% of the binder with respect to soil, as the previous test demonstrated that a high ratio is needed to get a consolidated product. This time, we also added SAP and water from the initial phase.
(a)3 g soil + 0.3 g Fe_2_(SO_4_)_3_ + 0.9 g binder(b)3 g soil + 0.3 g Fe_2_(SO_4_)_3_ + 0.9 g binder + 0.3 g SAP + 0.35 g water(c)3.5 g soil + 0.35 g Fe_2_(SO_4_)_3_ + 0.9 g binder(d)3.5 g soil + 0.35 g Fe_2_(SO_4_)_3_ + 0.9 g binder + 0.35 g SAP + 0.35 g water(e)4 g soil + 0.4 g Fe_2_(SO_4_)_3_ + 0.9 g binder

The prepared samples were treated as before, i.e., they were placed inside the SpaceQ at Martian conditions and frozen before the vacuum phase, and then, CO_2_ was injected and the table temperature was raised so that the product dries out and cures slowly. Once the RH was stabilized, the experiment was stopped, and the samples were taken out and weighed. Table 2 summarizes the comparison of the weights. The two samples shown in Figure 6b,d were mixed with SAP with 0.35 g of added water from the start. After 5 h of curation, these samples were cured into a solid block, and those with SAP and water have managed to hold 83% of the water added initially. 

The procedure, which has been here presented for illustration, allows to demonstrate on Earth within a controlled environment the production of a solid block under Martian conditions. This block retains water as part of its structural composition. As epoxy and SAP are not available on Mars, the process requires transporting them from Earth. This is always assumed as a base cost for all In Situ Resource Utilization methods for Mars. According to this specific study, we would need to transport 30% binder of the weight of the final product. In order to use these materials for construction, the structural integrity of these blocks should be demonstrated before. Future studies in the SpaceQ may explore other compositions, with other binders and other regolith simulants, concrete, etc. to evaluate its potential for construction, as suggested by other authors [28,31], and to optimize the process with respect to transportation of materials from Earth. The novelty of this material is that it has been cured completely in a low pressure, CO_2_-rich atmosphere with temperatures down to 258 K, whereas all studies done previously were cured under Earth ambient conditions. Furthermore, this is the first material that incorporates SAP and added water into its structural composition. Future experiments will be focused on characterizing the structural integrity and mechanical properties of materials cured under Martian conditions (i.e., tension, compression, thermal properties, and ionizing radiation screening properties). Similar experiments can be done to investigate the use of other binders, such as elemental sulfur (ES)-based binder, which have been suggested for Mars [28] or other Mars regolith simulants.

## 5. Instrument Testing in SpaceQ: HABIT GTS Operation

As a demonstration of the use of the chamber for instrument testing and calibration, here, we report the tests of the ground temperature sensor (GTS) of the Engineering Qualification Model (EQM) of HABIT (HabitAbility: Brines, Irradiation and Temperature) instrument onboard the ExoMars 2022 Surface Platform (ESA-IKI Roscosmos). This instrument will investigate the habitability of present-day Mars, monitoring temperature, winds, dust conductivity, ultraviolet radiation, and liquid water formation [1]. The GTS is a lightweight, low-power, and low-cost pyrometer that will measure the soil kinematic temperature of the Martian surface during the nominal mission lifetime of one Martian year. It benefits from a simple design with no moving parts and pointing to the Martian surface at an angle of 45°. The sensor acquires its heritage from the Rover Environmental Monitoring Station (REMS) ground temperature sensor (GTS), an instrument aboard the National Aeronautics and Space Administration (NASA) Mars Science Laboratory [37,38]. This section describes the calibration procedure of the 8–14 μm wavelength band-sensitive infrared thermopile and presents the results of the tests done inside SpaceQ (Figure 7). 

The SpaceQ provides appropriate atmospheric conditions in terms of temperature, pressure, and carbon dioxide atmosphere as that of Mars. Some of the instrument tests that can be carried out include instrument operation testing during thermal cycle in a vacuum and under the presence of Mars atmosphere, thermal balance studies and response to heaters, instrument material testing for outgassing, and similar science experiments relevant to the application area of instrument operation, to name a few. As an example, we demonstrate a part of the calibration of the ground temperature sensor (GTS) of the HABIT instrument. Other complementary calibration tests were performed outside of the chamber, using a blackbody source, whereas other tests were done using the Flight Model in a cleanroom facility [1]. 

Due to the test requirements, the cooled part of the setup was mounted in contact with the working table, and the rest was protected with an insulation layer. The SpaceQ is equipped with a DB-25 port on the inside of the chamber that is coupled with a similar connector on the exterior of the chamber via a vacuum sealed adapter. Hence, all the test equipment used inside the chamber must comply with this connection setup for power and data transmission (see Figure 2).

For the calibration tests, we used the Engineering Qualification Model (EQM) of HABIT, mounted on the SpaceQ’s working table with a spacer acting as an insulation layer and a thermographic paint-coated aluminum plate (similar to a blackbody) bolted directly on the working table. The objective of the calibration tests is to measure the response of the infrared thermopile of the GTS as a function of the temperature difference between the GTS ambient temperature and the target maintained at Mars relevant temperatures. Hence, the tests were carried out at temperatures between 10 and −50 °C and with a carbon dioxide pressure between 6 and 8 mbar. For reference, we used a MELEXIS MLX90614 infrared thermometer with a −70 to 380 °C temperature measurement range having a 0.5 °C measurement accuracy and 0.02 °C resolution.

Figure 8a shows the GTS response (in millivolts) as a function of the difference between the target temperature and GTS ambient temperature calibrated with the experimental setup in SpaceQ. 

The relation between the target temperature and the ground heat flux ($\Phi_{g}^{I}$) is shown in Figure 8b. The target temperature, T_g_ can then be calculated from the resultant ground heat flux using a second-order polynomial least-square fit as expressed in Equation (5).
(5)$$T_{g}=-0.0578.\Phi_{g}^{I}{}^{2}+2.0917.\Phi_{g}^{I}+276.2693$$

The retrieved target temperature is the brightness temperature considering the blackbody with an emissivity, ε = 1. For surfaces where ε ≠ 1, the assumed emissivity from the literature can be substituted to recalculate the $\Phi_{g}^{I}$. term. The accuracy of the temperature retrieval for a sample GTS measurement is shown in Figure 8c. The overall accuracy of temperature retrieval at a temperature between −50 and 10 °C is within 1.3 °C.

## 6. Conclusions

We have built a versatile environmental chamber to simulate Martian atmospheric conditions. The SpaceQ chamber is a unique facility that can operate with pressures from <10^−5^ mbar to ambient and temperatures from 163 to 423 K. As the SpaceQ has an internal volume of 27,000 cm^3^, it can expose multiple samples in one experiment. It can be used to qualify the behavior of specific components when exposed to thermal vacuum, outgassing, baking, low temperatures, and dry heat microbial reduction procedures. With its optical and electrical feedthroughs, it can log the data in real-time from the instruments and monitor the environmental parameters such as temperature, pressure, and relative humidity inside the chamber. Other existing simulating facilities have the possibilities to mimic planetary atmospheric composition, pressure, temperature, and irradiation. However, the SpaceQ chamber is specifically suitable to study the near-surface water cycle on Mars and can simulate the Martian diurnal/seasonal temperature and relative humidity variation within the Martian pressure range. This facility is more resourceful and can be used to test the curation of materials under the Martian environment and to study its degradation when exposed to the high-intensity UV lamp inside the chamber.

As an example of the tests that can be done, we have simulated (i) a plausible transition of night to day atmosphere-regolith water cycle on Mars, (ii) the curation of materials under dry Martian conditions, and (iii) the calibration tests of HABIT GTS. Future experiments will be focused on creating a complete diurnal cycle of a Martian solar day and simulating different regolith mixtures to check for brine stability, investigating the habitability of Martian soils, and studying different binder-based materials for concrete fabrication under Martian conditions.

Our studies demonstrate, as a first step, that it is potentially feasible to fabricate and cure under Martian conditions, within a few hours, small brick-like elements produced with a specific mixture of regolith, binders, and water retaining products. Future studies will evaluate the properties of these materials, testing for radiation screening, and their structural integrity to evaluate if the material can be further used as a construction material for Mars habitation.

Also, we shall study the phase characterization of these brines and minerals formed or altered inside the chamber using ultraviolet–visible–near infrared (UV–VNIR) spectrometry. This facility is unique for investigating the effect of water on the Martian surface. In particular, our studies have demonstrated that the regolith, when mixed with SAP, water, and binders and cured under Martian conditions, can retain more than 80% of the added water. 

## Figures and Tables

**Figure 1 sensors-20-03996-f001:**
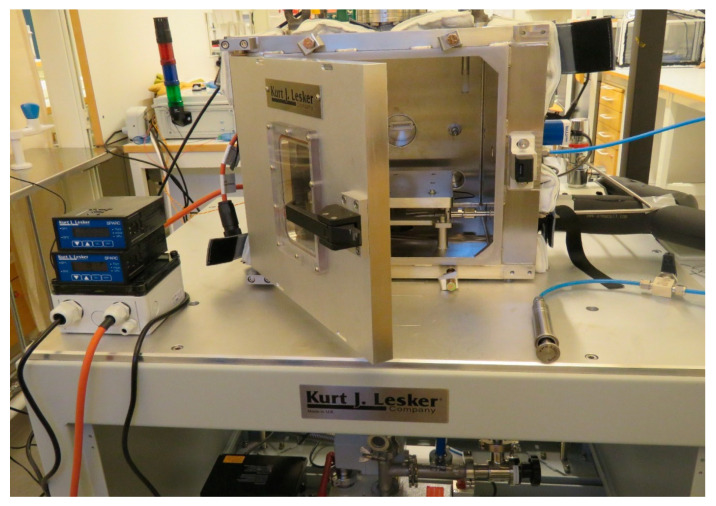
Overview of the SpaceQ chamber, without the heating jacket, fitted with different components.

**Figure 2 sensors-20-03996-f002:**
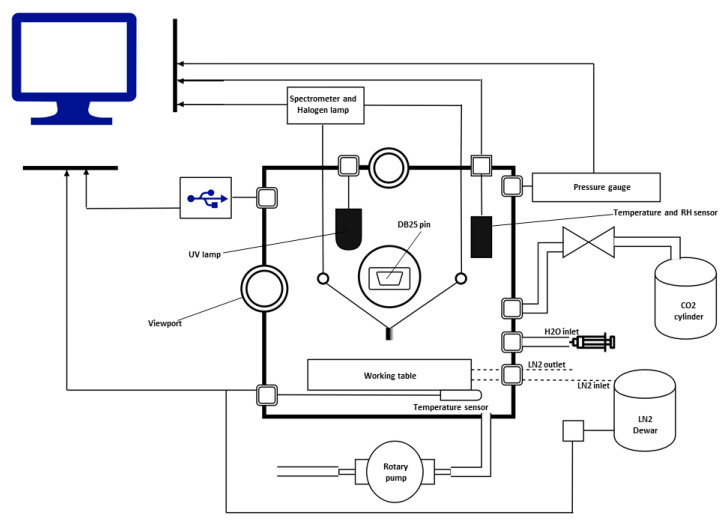
Schematic representation of SpaceQ chamber showing different components fitted.

**Figure 3 sensors-20-03996-f003:**
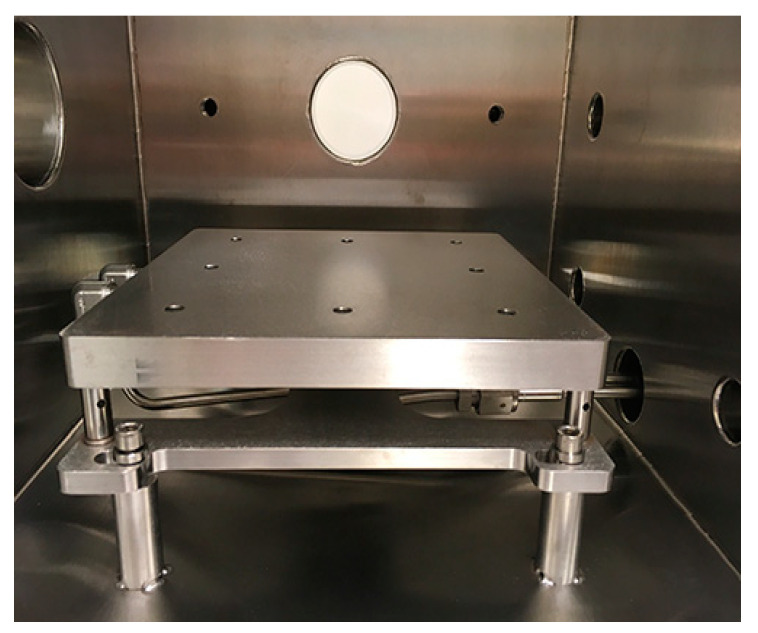
Detailed view of the cooling plate setup, which allows for the flow of liquid nitrogen.

**Figure 4 sensors-20-03996-f004:**
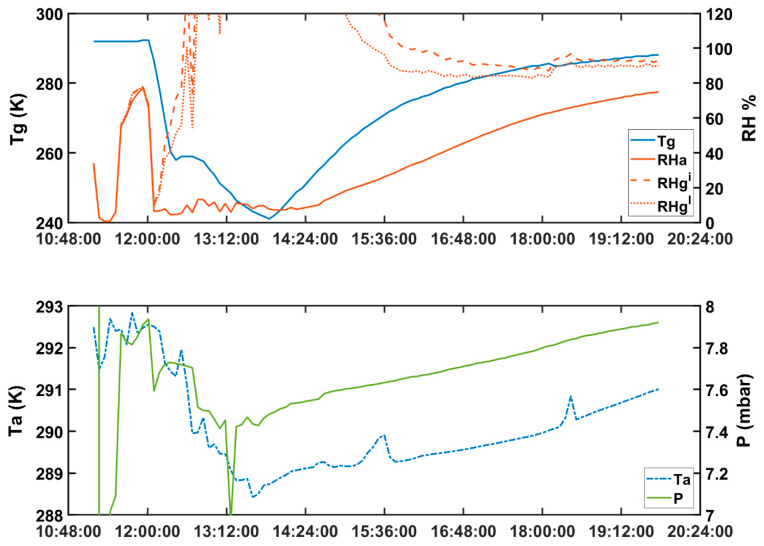
Example of the time evolution of some of the environmental parameters inside the chamber when simulating the transition from a Martian night to a Martian day. For this specific configuration, the ground relative humidity is saturated and allows for frost formation, when the ground temperature is under 260 K.

**Figure 5 sensors-20-03996-f005:**
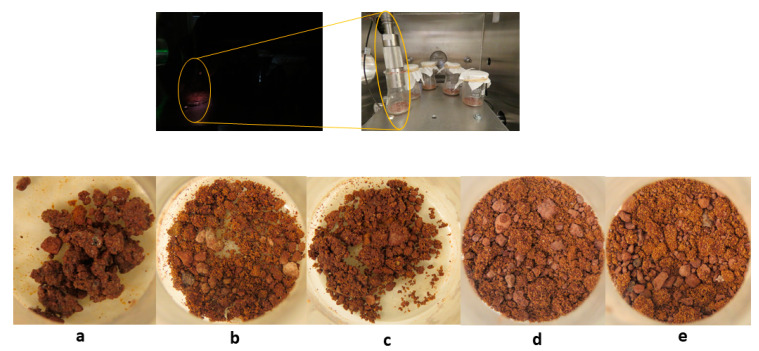
(**a**) 1 g soil, (**b**) 1.5 g soil with UV, (**c**) 1.5 g soil without UV, (**d**) 2 g soil, and (**e**) 2.5 g soil.

**Figure 6 sensors-20-03996-f006:**
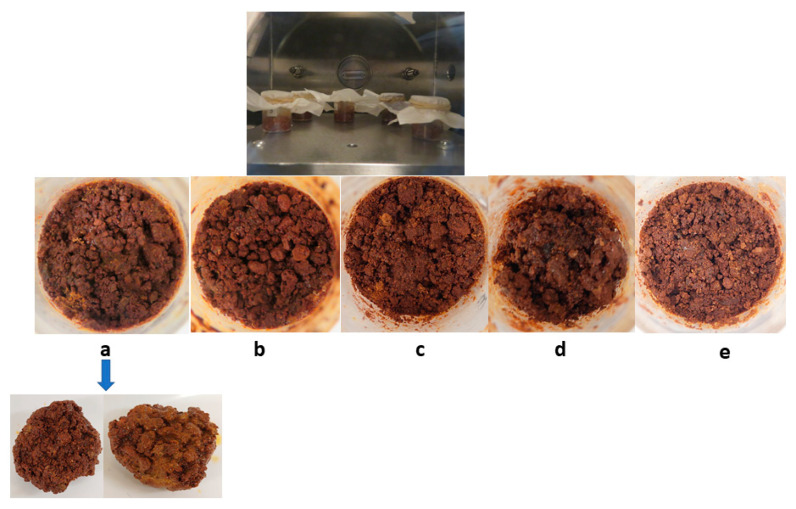
(**a**) 3 g soil, (**b**) 3 g soil + SAP, (**c**) 3.5 g soil, (**d**) 3.5 g soil + super absorbent polymer (SAP), and (**e**) 4 g soil. Examples of solid blocks of different amounts of soil and binder, sulfate, SAP, and water, after 5 h of a curation under Martian environmental conditions.

**Figure 7 sensors-20-03996-f007:**
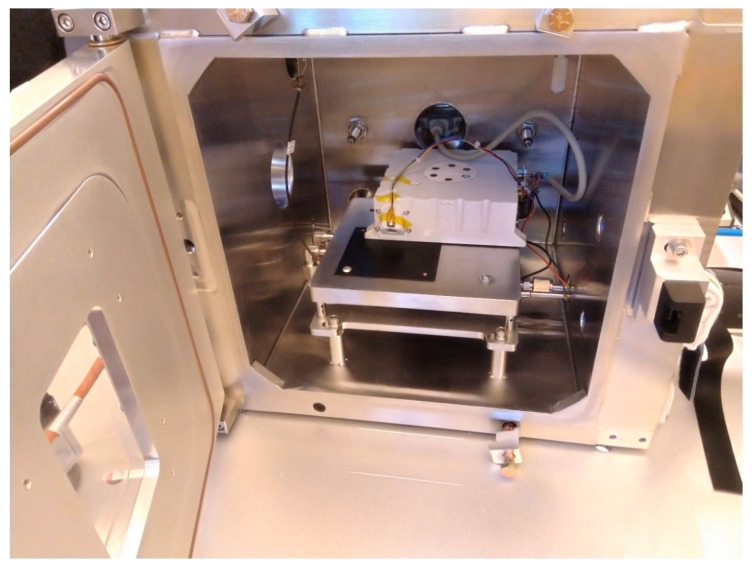
Configuration for the HabitAbility: Brines, Irradiation and Temperature (HABIT) ground temperature sensor (GTS) calibration inside SpaceQ chamber. The thermopile points to a small-sized thermographic paint-coated aluminum plate that serves as blackbody target, which is used as a reference for the calibration.

**Figure 8 sensors-20-03996-f008:**
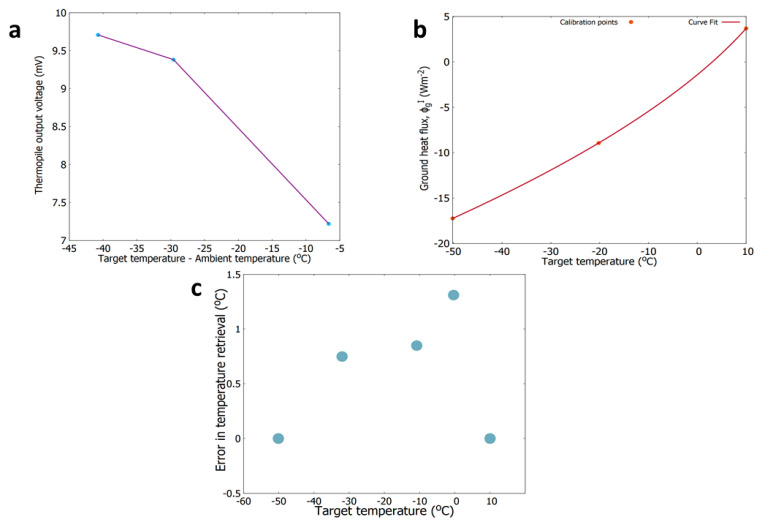
(**a**) Thermopile output voltage as a function of the difference between the target temperature and GTS ambient temperature. (**b**) Relationship between the target temperature and the ground heat flux as measured by the GTS. (**c**) Estimatn of error in temperature retrieval at different temperatures.

**Table 1 sensors-20-03996-t001:** Specifications of SpaceQ.

Parameter	Characteristics
Chamber dimensions	30 cm × 30 cm × 30 cm
Operating temperature	163–423 K
Operating pressure	< 10^−5^ to 1000 mbar
Viewport	Fused silica quartz
UV lamp	115–400 nm
Data output	USB and DB-25
RH and temperature	0–100% and 203–453 K
Gas inlet	CO_2_ and water
VNIR spectrometer	200–1100 nm

**Table 2 sensors-20-03996-t002:** Summary of the weights of samples.

Weight	Sample a	Sample b (SAP)	Sample c	Sample d (SAP)	Sample e
Initial (g)	4.2	4.85	4.75	5.45	5.30
Final (g)	4.18	4.79	4.73	5.39	5.27
Weight difference (g)	−0.02	−0.06	−0.02	−0.06	−0.03
Water retention%	n/a	83%	n/a	83%	n/a

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
