# Peer review of "Space Environmental Chamber for Planetary Studies"

_sensors, 2020, doi:10.3390/s20143996_

Round 1

Reviewer 1 Report

The paper by Ramachandran et al. describe a new developed versatile simulation chamber that operates under representative space condition: the SpaceQ chamber. This chamber can allow to test instrumentation, procedures and materials and evaluate their performance when exposed to outgassing, thermal vacuum, low temperatures, baking, dry heat microbial reduction. Authors show some applications of the SpaceQ chamber at simulated Martian conditions with and without atmospheric water.

I found this paper suitable for publication after a very few changes. Data presented might be helpful for astrobiological experiments to clarify the limit of habitability on Earth and explore extraterrestrial life.

Here I suggest some changes:

-Line 17: “Liquid Nitrogen” in “liquid nitrogen”.

-Line 30: add comma after regolith

-Line 31: change “be cured under Martian conditions” in “exposed to Martian conditions”

-Line 31: add comma after conditions

-Line 34: I suggest to keep Space along the text

-Line 42: add comma after calibrate

-Line 43: Change “It is the purpose...” in “The main purpose of this chamber is..”

-Line 49: change “to the surface of Mars” in “landing Martian surface”

-Line 53: “ultraviolet” in UV

Line 54: “during,” in “for”

-Lines 57-60: change “Since the 60s, there have been numerous developments of simulation chambers around the world. Starting from the chamber to study the behaviour of terrestrial microorganisms under artificial Martian  conditions [2], and continuing with multiple simulating facilities for planetary and space research [3-17].” in “Since the 60s, there have been numerous developments of simulation chambers around the world, including the chamber to study the behavior of terrestrial microorganisms under artificial Martian  conditions [2] and other multiple simulating facilities for planetary and space research [3-17].”

-Line 70: add comma after state.

-Line 90: 27000 in 27,000.

-Line 92: add space after 65.

-Line 190: add comma after injected.

-Line 224: am? pm?

-Line 229: put RHa in ()

-Line 253: please, be consistent along the text. Use always “figure” of “Figure”.

-Line 303: It should be “Figure 5.” or “Figure:”?

-Line 303: add dot after soil.

-Line 324: “Table:”?

-Line 328: “Figure 6:” or “Figure 6.?”

-Line 350: change “,” in “.” after calibration.

-Line 382: Figure.

-Line 403: 27000 in 27,000

-Line 417: delete “be”.

Author Response

Dear Reviewer, 

Thank you so much for your time in reviewing our manuscript. Please see the attachment for the response letter to your valuable feedback. 

Best regards,
Abhilash Vakkada Ramachandran et al. 

Reviewer 2 Report

Very interesting topic. Please see and consider below comments.

1. Please double check the grammar and spelling throughout the text.

e.g Figure 2. (typo) representation

e.g. line 412. (typo) Martian soil

2. Line 254. the goal of the experiment is to have a mixture to fabricate a solid block for construction on Mars. 

Line 299. The samples with a high binder to soil ratio form conglomerate. However, as shown in Figure 5, the samples are quite loose and dispersed. Please discuss the feasibility and physical meaning of the experimented mix as a construction material for Mars habitation.

4. Please explain in more detail of Figure 6. Sample (a) becomes solid? Does it have structural integrity for actual construction?

5. Please provide a comparison of the environmental chamber with similar products available on the market. What is the core novelty of the utilized chamber?

6. Researchers have developed sulfur based martian concrete. Is it possible to study the material in this chamber? Please discuss and elaborate.

Author Response

Dear Reviewer, 

Thank you so much for your time in reviewing our manuscript. Please see the attachment for the detailed response to your valuable feedback. 

Best regards,

Abhilash Vakkada Ramachandran et al.

Round 2

Reviewer 1 Report

Thank you to the authors for this revised version.

Reviewer 2 Report

The focus of the paper is the environmental chamber.

The utilized material does not seem structurally or economically feasible for martian construction. The selected binder - epoxy, is usually used as adhesive, glue or coating, but rarely as a binder for structural mass production. The structural integrity of the experimented material lacks of practical proof. Furthermore, epoxy cannot be easily produced on Mars. A binder-to-aggregate ratio as high as 30% would require transporting from earth to Mars high amount of epoxy for structural scale construction. 

It is suggested to modify the paper accordingly:

  • explain how and why the binder was selected
  • discuss and elaborate on the advantages/disadvantages of the tested material for martian construction 
  • provide comparison of the experimented material with the ones developed by other researchers, as cited in the paper
  • until further studies, remove the sections saying the cured material could be used for martian construction
  • focus on the functionalities of the chamber. 

Author Response

Dear Reviewer, 

Thank you for your time in reviewing our manuscript. Please see the attachment for a detailed response to your valuable comments. 

Best regards,

Abhilash 

Round 3

Reviewer 2 Report

The paper is suggested to be accepted as in the present form.